# Chronic Overexpression of Neuronal NRG1-III in Mice Causes Long-Term Detrimental Changes in Lower Motor Neurons, Neuromuscular Synapses and Motor Behaviour

**DOI:** 10.3390/ijms262311421

**Published:** 2025-11-26

**Authors:** Sara Salvany, Sara Hernández, Anna Casanovas, Sílvia Gras, Lídia Piedrafita, Mar Bosch-Queralt, Markus H. Schwab, Jordi Calderó, Josep E. Esquerda, Olga Tarabal

**Affiliations:** 1Grup de Patologia Neuromuscular Experimental, Departament de Medicina Experimental, Facultat de Medicina, Universitat de Lleida, 25198 Lleida, Spain; sara.salvany@udl.cat (S.S.); sara.hernandez@udl.cat (S.H.); anna.casanovas@udl.cat (A.C.); silvia.gras@udl.cat (S.G.); lidia.piedrafita@udl.cat (L.P.); jordi.caldero@udl.cat (J.C.); josep.esquerda@udl.cat (J.E.E.); 2Institut de Recerca Biomèdica de Lleida—Fundació Dr. Pifarré, IRBLleida, 25198 Lleida, Spain; 3Paul-Flechsig Institute for Brain Research, Medical Faculty, University Leipzig, 04103 Leipzig, Germany; mar.boschqueralt@medizin.uni-leipzig.de (M.B.-Q.); markus.schwab@medizin.uni-leipzig.de (M.H.S.)

**Keywords:** Neuregulin 1 (NRG1), neuromuscular junction (NMJ), plasticity, degeneration, synapse, motor neuron (MN), spinal cord, muscle

## Abstract

Neuregulins (NRGs) are ligands of tyrosine kinase receptors from the ErbB family and play multiple developmental roles. NRG1–ErbB signaling regulates myelination and has been associated with amyotrophic lateral sclerosis (ALS) pathology. Given the potential therapeutic relevance of this pathway for motor neuron (MN) diseases, we employed a transgenic (TG) mouse with persistent neuronal overexpression of neuregulin type III (NRG1-III) to investigate its impact on the neuromuscular system. We performed an analysis of phenotypic changes in this TG model, including motor behavior, neuropathological evaluation by immunocytochemistry and ultrastructural examination of the spinal cord, peripheral nerves, and neuromuscular junctions (NMJs). Calcium dynamics in cultured MNs were also examined. We found that cholinergic C-boutons on TG MNs, where NRG1-III typically accumulates, exhibited upregulation of C-bouton-associated proteins and expansion of the subsynaptic cistern (SSC)-associated endoplasmic reticulum. Calcium imaging revealed altered homeostasis in TG MNs, accompanied by the upregulation of molecules linked to axonal plasticity. At NMJs, regressive changes involving autophagic dysregulation were observed. These alterations were accompanied by increased motor activity in behavioral tests. Overall, our findings indicate that persistently elevated NRG1-III signaling compromises MN connectivity and long-term health, a factor to consider when developing therapeutic strategies for neurodegenerative diseases such as ALS.

## 1. Introduction

Neuregulins (NRGs) are neurotrophic factors that play a crucial role in cell communication and differentiation in a variety of organs. The NRG family comprises six genes, which give rise to multiple isoforms through alternative splicing. NRG-encoded proteins are structurally related to the epidermal growth factor (EGF) family and serve as ligands for the ErbB family of tyrosine kinase receptors (reviewed in [1,2]). NRG1, the most extensively studied family member, regulates key developmental processes such as synapse formation and myelination in both the peripheral and the central nervous system (CNS) [3,4,5]. Previously, we showed that, in α-motor neurons (MNs), NRG1 accumulates at the postsynapse of cholinergic afferents known as C-boutons [6]. These large synapses, which originate from cholinergic V0C interneurons, are characterized by an ER-derived structure known as the subsynaptic cistern (SSC) that is closely apposed to the postsynaptic membrane [7]. In addition to NRG1, C-boutons contain several signaling molecules organized into non-overlapping microdomains at both the postsynaptic membrane and the SSC. These include the M2 muscarinic acetylcholine receptor (M2), voltage-gated K^+^ channel 2.1 (Kv2.1), Ca^2+^-activated K^+^ channel, the sigma-1 receptor and phospho-c-jun Y172-like antigen [8,9,10,11,12]. We also observed expression of the phosphorylated form of the NRG1 receptor ErbB2 in the presynaptic compartment of C-boutons, suggesting a trans-synaptic NRG1-ErbB signaling mechanism [6,13]. C-boutons play a major role in modulating MN excitability by regulating K^+^ currents during afterhyperpolarization. However, the precise functions of C-bouton proteins in regulating MN properties remain largely unexplored [14]. Following axonal injury, MNs undergo significant changes that result in the loss of afferent synapses, including C-boutons, and in the local recruitment of microglia [13,15]. We previously showed that, after axotomy, the SSC-associated protein architecture, including NRG1, is disrupted, and recruited microglia exhibit positive chemotaxis towards C-boutons [13,16].

NRG1 isoforms are grouped into types I, II, and III (NRG1-I, -II, and -III), with NRG1-III being the most abundant isoform in α-MNs [6,17]. Proteolytic cleavage of its extracellular domain by the protease BACE1 results in the EGF-like domain remaining anchored to the plasma membrane, unlike NRG1-I and -II, where cleavage releases the EGF-like domain. Differential processing gives rise to distinct signaling modes: paracrine signaling for NRG1-I and -II and juxtacrine signaling for NRG1-III [18]. To investigate the functional implications of different NRG1 isoforms in the nervous system, transgenic (TG) mouse models overexpressing specific isoforms have been generated. Studies on these TG lines have revealed that NRG1-III is a key regulator of Schwann cell differentiation and myelination during peripheral nerve development and repair after injury [19,20,21]. At the neuromuscular junctions (NMJs), NRG1-III overexpression activates terminal Schwann cells (tSCs), promoting terminal motor axon plasticity beyond the normal period of postnatal remodeling [22]. Within the CNS, NRG1 plays a role in the establishment of cortical circuitry, balancing excitatory and inhibitory neurotransmission, and modulating synaptic plasticity [23]. We previously demonstrated that different NRG1 isoforms exert distinct effects on the organization of C-bouton components: NRG1-III acts as an organizer of ER-plasma membrane (ER-PM) contacts resembling postsynaptic SSC structures, while NRG1-I promotes enlargement of vesicular acetylcholine transporter (VAChT)-containing presynaptic C-bouton-like structures [16].

NRG1 has been implicated in pathogenic mechanisms of several neurodegenerative diseases, including amyotrophic lateral sclerosis (ALS). Mutations of the human *ERBB4* gene, which encodes the main NRG1 receptor in the CNS, have been identified as contributors to familial ALS [24]. Furthermore, both molecular and structural alterations in C-boutons have been reported in ALS and spinal muscular atrophy (SMA) during disease progression [6,25,26,27,28]. However, studies investigating the potential role of NRG1 in neuropathological conditions have yielded conflicting results. While viral-mediated delivery of NRG1-III into the spinal cord of ALS mice restored C-bouton numbers and increased survival [29,30], intraventricular administration of NRG1 showed no therapeutic benefit whereas neuroprotective effects have been reported following blockade of endogenous NRG1 signaling [31]. Recently, we examined NRG1-III neuronal overexpression in the SOD1 mouse model of ALS, showing that NRG1-III overexpression accelerates disease onset, worsens motor phenotypes and induces astrogliosis, but has no effect on the lifespan of ALS mice [32]. Moreover, boosting NRG1-III expression in SMA mice did not prevent distal axon degeneration nor motor behavior or survival [33]. Thus, the precise role of NRG1-ErbB signaling in neurodegenerative diseases remains largely undefined. Further elucidation of the involvement of this signaling module in the neuromuscular system is critical for the development of future therapies targeting NRG1-ErbB pathway in MN diseases.

Here, given its emergence of NRG1-III as a potential therapeutic target, we investigated the role of NRG1-III in the neuromuscular system through a comprehensive analysis of MN central synapses, motor axons, and NMJs in TG mice with neuronal NRG1-III overexpression. Our results revealed several alterations in these compartments, ultimately culminating in degenerative processes at the NMJs.

## 2. Results

### 2.1. Neuregulin Type III (NRG1-III) Overexpression in Mice Increases Motor Activity and Alters Ventral Root Axon Myelination

We first examined the impact of NRG1-III overexpression in MN, beginning at perinatal stages on motor performance and lifespan. No significant differences in survival were observed between TG mice overexpressing NRG1-III and wild-type (WT). This was consistent across sexes (WT males: 702.5 ± 21.5 days, *n* = 4; WT females: 699.7 ± 46.5 days, *n* = 4; TG males: 687.7 ± 144.8 days, *n* = 12; TG females: 748.42 ± 90.2, *n* = 12; one-way Anova). Motor behavior was assessed using the Open-field test. Total distance covered and average velocity were measured at different time points from middle-aged to old-aged mice (330–740 days). TG mice showed increased distance covered and higher average velocity compared to WT, with significant differences between 380 and 560 days, consistent with previous data observed at younger stages [34] (Figure 1A,B). The CatWalk system revealed reduced paw intensity in TG mice compared to WT across most time points analyzed (Appendix A). These results suggest that neuronal NRG1-III overexpression initially enhances locomotor activity, but this hyperactivity diminishes in aged stages (after 590 days). At this stage, age-dependent neurodegeneration may counteract the NRG1-dependent hyperactive phenotype. Overall, these data are in line with motor behavior analyses performed at earlier ages [34].

NRG1 is a key regulator of developmental myelination. Overexpression of NRG1-III in MNs of TG mice results in a well-documented phenotype of hypermyelination [20,21], which was confirmed here (Figure 1C). Additionally, we explored whether NRG1-III overexpression retains the capacity to induce long-term hypermyelination in old mice. We determined the G-ratio in L4 ventral roots (VRs), which contain the proximal segments of motor axons that project into the sciatic nerve. Compared to VRs of adult WT mice, those of adult TG animals exhibited a reduced G-ratio, indicating hypermyelination. In old TG mice, however, we observed a decrease in axon myelination in old TG mice (Figure 1C–F, Appendix A). The total number of axons remained unchanged (Figure 1G), but we observed a tendency towards higher numbers of degenerating axons in VRs of both adult (*p* = 0.25) and old (*p* = 0.22) TG mice (Figure 1C,H). Since we detected no significant axonal loss in old nerve roots, it is plausible that a compensatory mechanism, such as collateral axonal sprouting from healthy axons, occurred. In fact, when comparing histograms of axonal diameter distribution in adult and old TG L4 VR, an increased frequency of small-diameter axons (<8 µm) was observed in old TG mice. In addition, a small increase in the proportion of very-large-diameter axons was detected in TG mice, which likely represents a population of swollen, degenerating axons (Figure 1I).

### 2.2. Impact of Long-Term NRG1-III Overexpression on Motor Neuron (MN) Synaptic Afferents and C-Bouton Organization

NRG1 is prominently expressed in the soma and proximal dendrites of α-MNs, in which it is concentrated adjacent to VAChT-positive presynaptic terminals in the postsynaptic compartment of cholinergic C-bouton synapses [6,13]. We previously showed that overexpression of NRG1-III in adult TG mice leads to widespread SSC accumulation of NRG1-III in MN somata, which display a set of SSC-specific molecular constituents [16]. However, the long-term consequences of these changes on MNs have not yet been explored [6,16]. Thus, we investigated the impact of NRG1-III overexpression on MN biology from embryonic development to ageing, with a particular focus on plastic changes in C-boutons and other afferent synapses to MNs. Measurements of MN soma area using Nissl staining revealed a similar increase in WT and TG mice from birth to postnatal stages (P10). In adult mice (P60), MN soma size was significantly larger in TG than in WT mice. This was not observed in old animals (P600 [Figure 2A]). Moreover, no changes were observed in the total number of L4 ventral root axons, suggesting unaltered MN numbers in TG mice, (Figure 1G).

NRG1 immunostaining revealed that NRG1 expression in MNs increased from P10 to adulthood in both WT and TG mice, but this elevation was more pronounced in TG mice. In line with continuous Thy1.2-promoter activity, the expected decline of NRG1 during aging was not observed in TG animals (Figure 2B,D). Additionally, we analyzed hemagglutinin (HA)-tag immunolabeling to determine whether NRG1-III overexpression in spinal cord MNs of TG mice colocalized with pan-NRG1 immunolabeling. A match between the two markers was observed at the periphery of MN somata (Figure 2D). Despite postsynaptic NRG1-III overexpression, the developmental profile of VAChT was similar in WT and TG mice (Figure 2C,D). Moreover, we confirmed co-expression of p-ErbB2 in the presynaptic compartment of C-boutons, overlapping with VAChT [13]. (Figure 2E) and found no differences in TG mice (Figure 2F–M). Ultrastructural analysis revealed that, as expected, in control MNs, SSCs are confined to postsynaptic sites facing C-bouton terminals (Figure 3A,B). NRG1-III overexpression promotes SSC biogenesis, extending its domain beyond the postsynaptic region (Figure 3C,D). Notably, SSC enlargement remains stable in old TG mice (Figure 3E,F). Collectively, these findings are consistent with a specific role of NRG1-III in organizing the postsynaptic, but not the presynaptic compartment of C-boutons [16].

Finally, we investigated the impact of NRG1-III overexpression on MN synaptic connectivity beyond cholinergic inputs. Quantification of excitatory synapses using antibodies against vesicular glutamate transporter 1 (VGluT1) and vesicular glutamate transporter 2 (VGluT2) revealed a higher density of positive spots on the MN surface of TG mice (Figure 3G–I). In contrast, quantification of inhibitory synapses using antibodies against vesicular GABA transporter (VGAT) showed no significant differences in synaptic density between TG and WT MNs. (Figure 3G,J). These observations suggest that NRG1-III overexpression in either MNs or sensory neurons specifically results in increased glutamatergic connectivity in MNs.

### 2.3. Sustained NRG1-III Overexpression Promotes MN Plasticity and the Development of a Fast-Fatigable MN Phenotype

Calcitonin gene-related peptide (CGRP), a member of the calcitonin family of peptides [35], is produced in peripheral and central neurons and is associated with axonal growth and synaptic plasticity, particularly in NMJs [36,37,38]. CGRP immunoreactivity was increased in MNs from adult TG mice compared to WT (Figure 4A,B). Notably, during aging, the CGRP expression was further increased in WT. Conversely, CGRP immunoreactivity was reduced in MNs of old TG mice when compared to both old WT and adult TG MNs. These findings suggest that MNs of old TG mice may be less responsive to the reactive plastic changes mediated by CGRP upregulation, which occurs in old WT MNs [39]. A similar pattern was observed for matrix metalloproteinase-9 (MMP9) immunoreactivity, with an increase in adult TG MNs followed by a subsequent decrease in old TG MNs (Figure 4A,C). MMP9 is specifically expressed by fast MNs, which are selectively vulnerable in MN diseases [40]. Moreover, old mice exhibited a higher degree of CGRP-MMP9 colocalization compared to adult mice, although no differences were observed between MNs from old WT and TG mice (Figure 4A,D). Furthermore, we have not observed differences in the degree of MMP9+ MNs, HA+ MNs, or double/triple-labeled MNs (CGRP+/HA+, MMP9+/HA+, and CGRP+/MMP9+/HA+) between adult and old TG mice.

Immunoglobulin heavy chain binding protein (BiP), an ER stress sensor associated with MN degeneration, was also examined [41,42,43]. A significant increase in BiP intensity was observed in MN from adult TG mice compared to adult WT animals. This suggests that NRG1 overexpression may lead to increased ER stress, potentially linked to excessive SSC-like ER biogenesis (Figure 4E,F).

### 2.4. Axotomized MNs in Transgenic Mice Exhibit Disrupted NRG1 Compartmentalization and Exacerbated Microglial Recruitment

Peripheral nerve transection alters the soma of MNs and surrounding glial cells, a process aimed at restoring MN connectivity with their targets [16,44,45]. We previously reported that increased microglial recruitment to axotomized MNs involves selective tropism for contacting afferent synaptic terminals, particularly those with glutamatergic and cholinergic phenotype (C-boutons) [13]. Thus, C-bouton-associated proteins may exert a chemoattractive signal for microglia. Moreover, NRG1-III overexpression-linked SSC expansion in TG animals is accompanied by a corresponding enrichment of additional C-bouton-associated proteins [16]. Therefore, we explored how altered molecular organization of the MN surface affects microglial response post-axotomy. When we measured the MN surface covered by ionized Ca^2+^-binding adaptor molecule 1 (Iba1)-labeled microglia in adult mice 7 days post-axotomy, we found an exacerbation of microgliosis in TG mice compared to WT (Figure 5A,C,E,G,I). However, loss of cholinergic afferent boutons following axotomy (measured by MN surface area covered by VAChT-positive synaptic profiles) was similar in TG and WT mice (Figure 5B,D,F,H,J). Next, we measured the extension of NRG1 profiles closely associated with MN surface. Axotomized MNs of WT mice showed a tendency (*p* value = 0.78) to a reduction in NRG1 immunolabelling compared to uninjured controls. In contrast, axotomized MNs of TG mice exhibited a significant and pronounced decrease in NRG1 expression relative to TG controls (Figure 5B,D,F,H,K). These results demonstrate that NRG1-III overexpression has no effect on axotomy-induced C-bouton loss. However, in contrast to WT animals, axonal injury in TG mice leads to a prominent reduction in surface-associated NRG1 clusters, suggesting a higher susceptibility of TG NRG1 clusters to disruption in response to injury.

### 2.5. In Vitro Expression of C-Bouton-Associated Molecules in MNs from NRG1-III Overexpressing Mice

Cultures from whole-dissociated spinal cords of WT and NRG1-III TG mice were maintained from 14 to 23 days in vitro (DIV), during which astrocytes formed an almost continuous layer that supported the growth of various types of neurons, including MNs. MNs were identified based on their size, shape, and choline acetyltransferase (ChAT) immunolabeling. We then explored the distribution of NRG1 immunolabeling in MNs within these cultures. NRG1 clusters were scattered across the MN surface in both WT and TG MNs, resembling the pattern observed in vivo in WT animals (Figure 6A,B). Additionally, the soma of some TG MNs exhibited an extended peripheral accumulation of NRG1, which colocalized with HA and mirrored the pattern found in TG mice in vivo (Figure 6A,C). Next, we explored the in vitro expression of other well-characterized C-bouton proteins, such as Kv2.1, M2, and VAChT [9,11] (Appendix A). In TG MNs, increased peripheral Kv2.1 immunolabeling was observed alongside NRG1, although the overlap was not complete (Appendix A); M2 exhibited a similar distribution pattern to NRG1, appearing as scattered spots near the MN surface in some cells and as enlarged peripheral labeling in others (Appendix A). While multiple synapses on MN soma and neurites were labeled with synaptophysin (SYN), no colocalization was observed between either NRG1 or M2 and SYN, suggesting that these proteins accumulate in MNs outside of synaptic sites (Appendix A). Remarkably, VAChT-positive synaptic boutons were scarce in vitro. In some cases, VAChT-positive puncta were heterogeneously distributed along neurites and on the MN surface; however, in these cases, VAChT immunolabeling did not usually colocalize with NRG1 (see Appendix A). These findings suggest that NRG1 clusters in vitro do not form fully differentiated C-boutons comparable to those observed in vivo.

### 2.6. In Vitro Development of Excitatory Inputs to MNs from NRG1-III-Overexpressing Mice and Changes in Glutamate Receptor-Mediated Vulnerability

Given the observed excessive number of excitatory synaptic inputs on spinal cord MNs in vivo, we sought to determine whether similar alterations would occur in vitro. We assessed the density of VGluT1-immunolabeled spots on neurites and observed higher density in TG cultures, consistent with in vivo findings (Figure 6D,E). In contrast, the more general synaptic marker SYN did not show differences between WT and TG cultures, indicating that synapses are not globally affected by NRG1-III overexpression. In conclusion, findings in cultured MN (in the absence of sensory neurons with NRG1-III overexpression) suggest a MN-intrinsic regulation of glutamatergic synapses.

NRG signaling has been involved in the regulation of N-Methyl-D-aspartate acid (NMDA) receptors in cortical interneurons. Specifically, activation of NMDA receptors triggers the shedding of NRG2 extracellular domain, which promotes ErbB4 association with NMDA receptors. This association leads to the rapid internalization of surface receptors and potent downregulation of NMDA receptor currents, but does not affect AMPA receptor currents [46]. We explored whether a similar interplay could also occur in spinal cord cultures overexpressing NRG1-III. We first investigated whether NRG1-III overexpression affects MN survival. Quantification of MN numbers revealed a decrease in MN survival in TG cultures (Appendix A), suggesting that NRG1-III overexpression has detrimental effects on MNs. Next, we assessed MN death following acute NMDA application for 30 min. We observed a trend (*p* value = 0.0562) towards decreased MN death in TG cultures compared to WT (Appendix A), suggesting changes in NMDA receptor expression in TG cultures. Thus, we examined NMDA receptor levels in WT and TG cultures by Western blotting. A trend toward reduced levels of the NMDA receptor subunits GluN2A (*p* = 0.2), GluN2B (*p* = 0.3), and GluN1 (*p* = 0.4) was observed in TG cultures (Appendix A). Moreover, immunocytochemistry against the glutamate receptor subunit GluN2B and the AMPA receptor subunit GluA2 revealed a punctate pattern in the soma of WT and TG MNs, with some MNs displaying a more robust surface labeling (Appendix A). Quantification revealed lower expression of GluN2B but no change in GluA2 levels in TG cultures (Appendix A). Taken together, our data suggest that NRG1-III overexpressing MNs are less susceptible to excitotoxicity during early stages, potentially due to reduced NMDA receptor expression, suggesting that mechanisms other than glutamate- mediated excitotoxicity are involved in the unhealthy state of MNs seen after NRG1 overexpression.

### 2.7. Dysregulation of Calcium Homeostasis in MNs from NRG1-III Overexpressing Mice

Ca^2+^ signaling plays a key role in regulating neuronal excitability. In MNs, C-boutons are important regulators of excitability by modulating K^+^ current-mediated afterhyperpolarization [47]. Furthermore, intracellular Ca^2+^ dynamics at SSC influence the activity of Ca^2+^-dependent K^+^ channels, which are also spatially enriched at C-boutons [8]. While we were unable to observe full development of C-bouton-like synapses on MNs in vitro, clusters of molecules typically found at the postsynapse of C-boutons were present in these cultures and were altered by NRG1-III overexpression. Hence, we used this in vitro system to explore the impact of NRG1-III overexpression on Ca^2+^ homeostasis using fura-2 imaging in cultured MNs. Measurements of basal Ca^2+^ levels revealed higher levels in TG MNs from 19 to 22 DIV (Figure 7A). Notably, we observed spontaneous MN activity in both WT and TG cultures, but TG exhibited more frequent rhythmic patterns compared to WT MNs (Figure 7B,C). No differences in the amplitude of Ca^2+^ transients were detected between the two groups (Figure 7B,D).

To investigate cytoplasmic-ER Ca^2+^ signaling after ER replenishment, we stimulated cells with caffeine, a ryanodine receptor agonist and potent Ca^2+^ mobilizer from ER stores [48]. After washing, thapsigargin was applied; to selectively inhibit the ER-associated Ca^2+^ pump that induces irreversible depletion of ER Ca^2+^ stores [49]. Measurements of cytoplasmic Ca^2+^ signals revealed no impact of NRG1-III overexpression on ER Ca^2+^ dynamics in this paradigm (Appendix A). We also assessed Ca^2+^ influx into MNs following application of the glutamate receptor agonists NMDA and kainic acid (KA) but observed no differences between WT and TG cultures (Appendix A). Additionally, following KCl-induced depolarization, Ca^2+^ transients remained unchanged in TG cultures (Appendix A). In summary, the Ca^2+^ imaging experiments revealed subtle alterations in Ca^2+^ signaling, including slightly elevated basal Ca^2+^ levels and more rapid spontaneous transients. These findings suggest that NRG1-III overexpression may induce dysregulation of Ca^2+^ homeostasis, potentially contributing to changes in MN activity. Overall, our results from Ca^2+^ dynamics indicate that PM instead of ER-SSC accounts for the increased spontaneous activity observed in TG MNs.

### 2.8. Overexpression of NRG1 in MNs Leads to Plastic Changes in Neuromuscular Junctions

Morphological alterations in NMJs induced by NRG1-III overexpression were examined in two different hindlimb muscles tibialis anterior (TA) and soleus (SOL) from adult and old mice, thereby extending previous findings in NRG1-III TG mice obtained at postnatal and young adult ages [22]. These muscles were chosen for their distinct fiber composition: the SOL muscle predominantly contains slow-twitch fibers, whereas the TA muscle has a higher proportion of fast-twitch fibers. We performed immunostaining using 68kDa neurofilament-L (NF68) and synaptic vesicle glycoprotein 2A (SV2) to visualize nerve terminals and combined it with α-bungarotoxin (α-Bgtx) to label acetylcholine receptors (AChRs) at the postsynaptic NMJ membrane. No significant changes were observed in the total size of motor endplates in TG compared to WT muscles. However, NMJs in adult SOL muscles were smaller compared to those in adult TA muscles and the NMJs from old TA muscles were smaller than those in adult (Figure 8A). Additionally, adult TG muscles displayed a significant increase in NMJ perimeter (Figure 8B). Next, we measured the postsynaptic α-Bgtx-labeled area. TG TA muscles exhibited smaller areas compared to WT TA muscles in both adult and old mice, but no differences were observed in the size of SOL muscles between TG and WT (Figure 8C). Small clusters of extrasynaptic AChRs scattered along muscle fibers, characteristic of denervated fibers, have been previously observed in muscles of old mice [39,50]. Consistent with the presence of denervated fibers, TA muscles of old TG mice exhibited a higher number of AChR-dispersed spots compared to WT, while no differences were found in SOL muscles between both groups (Figure 8D). Furthermore, changes in NMJ morphology from a well-defined pretzel-like shape to a more fragmented appearance, have been reported in old muscles and NRG1-III overexpressing adult muscles [22,39,51]. Consistent with these reports, we observed an increase in the number of NMJ fragments in TG muscles of adult mice, and this increase was maintained in old TA muscles, but not in old SOL muscles (Figure 8E). The number of fragmented NMJs also significantly increased in TG adult TA muscles, while it only tendentially increased in adult TG SOL muscles (*p* value = 0.16) (Figure 8F). To explore whether NRG1-III overexpression induces changes in NMJ innervation patterns, we analyzed the colocalization of postsynaptic α-Bgtx with presynaptic nerve terminal labeling. TG mice exhibited increased partial denervation in TA muscles, suggesting a greater loss of nerve terminal branches at these NMJs (Figure 8G). In muscles from old animals, this denervation was accompanied by some NMJs displaying morphologies indicative of degeneration (Figure 8M,O). Notably, we observed an increase in endplate degeneration of TG TA muscles. Endplate degeneration of TG SOL muscles was also tendentially higher in TG mice (*p* value = 0.22) (Figure 8H). These results suggest that persistent NRG1-III overexpression in MNs triggers regressive changes at the NMJs, which may ultimately lead to their degeneration in old mice.

As NRG1 promotes Schwann cell proliferation, including tSCs at NMJs [22], we explored whether NRG1-III overexpression causes persistent changes in tSCs in adult and old mice. We performed tSC immunolabeling with S-100 antibody combined with 4′,6-diamidino-2-phenylindole dihydrochloride (DAPI) for nuclear labeling. In adult TG mice, a higher number of tSCs was observed in SOL NMJs compared to WT animals, as previously described [22]. However, no differences were detected in old mice. These results suggest that the capacity of NRG1-III to induce tSCs proliferation is limited in old mice and associated with other major alterations at the NMJs (Figure 8I,Q–T).

CGRP is expressed in MNs and is axonally transported to the NMJ. CGRP levels increase during NMJ development and decrease in mature NMJs. However, when axonal growth and plasticity are induced in nerve terminals, CGRP levels are again elevated [36,37,38]. Consistent with the plastic changes observed in NMJs of NRG1-III overexpressing mice, adult TG mice exhibited high levels of CGRP. In contrast, NMJs from old TG mice displayed CGRP levels similar to those found in WT mice (Figure 9A,B). In addition to CGRP, growth-associated protein 43 (GAP-43) expression is closely linked to axonal growth and NMJ plasticity [52]. To explore whether alterations in NMJ morphology were accompanied by changes in GAP-43 levels, we determined the expression of this protein in adult and old mice. SOL muscles of adult and old TG mice showed higher levels of GAP-43, while no differences were observed in TA muscles (Figure 9C,D). These results suggest that axonal NRG1-III overexpression in adult SOL NMJs, likely through increased NRG1-III-tSC signaling, induces nerve terminal plasticity regulated by CGRP and GAP-43. Overall, these data demonstrate a differential reaction of slow- vs. fast-twitch muscles in TG mice.

Ultrastructural examination of old TG NMJs revealed significant alterations consistent with the degenerative changes observed in confocal images. While some NMJs retained normally appearing nerve terminals (Figure 10A,C), others showed an accumulation of multilamellar and electron-dense structures, most likely representing endosomal/autophagic/lysosome-like vacuolar elements and multivesicular bodies (Figure 10B,D). These features indicate ongoing degradation of nerve terminals involving enhanced autophagic activity. Additionally, in some areas of the muscle surface, we identified remnants of former NMJs, characterized by remaining postsynaptic folds in the muscle membrane covered by tSCs, but unmatched by presynaptic elements. These findings suggest nerve terminal retraction and muscle denervation (Figure 10E). The increase in tSC processes reported during TG NMJ development [22] seems to be a transient phenomenon, as this was not detected at more advanced ages.

## 3. Discussion

Although NRG1 is a widely studied pleiotropic growth factor, its role in the neuromuscular system remains poorly understood. In this study, we confirmed structural changes in C-bouton-associated ER-PM contacts in TG mice overexpressing NRG1-III. In addition, we investigated the long-term organization of C-boutons and other synaptic inputs. Furthermore, we explored the effects of these changes on NMJ plasticity and motor behavior in TG animals.

NRG1 is essential for axonal myelination through the activation of ErbB receptors on Schwann cells [5,23]. Consistent with this role, we observed hypermyelination in the ventral axons of adult TG-NRG1-III mice. However, this effect was not observed in aged mice, where motor axons displayed increased susceptibility to degeneration. These findings suggest that chronic overexpression of NRG1 has detrimental effects on axonal integrity over time. It is interesting to note that in several pathological models, hypermyelination precedes hypomyelination and axonal pathology [53,54]. This aligns with the worsening effects of NRG1-III overexpression on the neuromuscular system in the context of MN disease [32] and the reduced cell survival we observed in cultured MNs. Finally, NRG1-III overexpression increased locomotor activity, indicative of alterations MN excitability and enhanced axonal firing. In addition to an imbalance of excitatory inputs, intrinsic factors within peripheral nerve may also contribute to the enhancement of neuromuscular electrical activity. On this line, it has been reported that NRG1-mediated hypermyelination also leads to increased sympathetic nerve excitability which is a key factor in the pathogenesis of primary palmar hyperhidrosis [55].

NRG1 is a transmembrane protein component of C-bouton-associated SSCs [6,17]. In our previous work, we demonstrated a marked expansion of SSC-like membranes in MN of NRG1-III TG mice in correlation with an enlargement of NRG1 labeling near the MN surface, while the presynaptic component of cholinergic C-boutons remained unaltered [16]. Consistent with these findings, we demonstrate that the age-dependent decline in the density of cholinergic bouton afferents on MNs is not altered by NRG1-III overexpression. Moreover, the distribution of the NRG1 receptor pErbB2 in the presynaptic compartment of C-boutons remains unchanged in TG mice. We extended our analysis to spinal cord-derived MN cultures from TG mice. NRG1 was broadly distributed along the periphery of a MN subpopulation, resembling the in vivo pattern. Similarly, postsynaptic C-bouton proteins, such as Kv2.1 and M2 exhibited an extended distribution, suggesting that increased NRG1 affects a wider set of postsynaptic and SSC-associated proteins. However, full C-bouton development, including proper alignment of pre- and postsynaptic structures, was not observed. Despite abundant SYN-labeled boutons, VAChT-positive cholinergic terminals were rare. Thus, while MNs express postsynaptic C-bouton proteins in vitro, they lack their presynaptic counterparts, even under NRG1-III overexpression. This mismatch might reflect the immature developmental stage of cultured MNs or the insufficient in vitro development of V0C interneurons, the primary source of C-boutons in vivo.

MN firing is regulated by a balance of excitatory and inhibitory synapses, with C-boutons playing a pivotal role in regulating α-MN excitability during locomotor behavior [7,14]. As we observed no differences in C-bouton synapse density in TG MNs, we investigated whether NRG1-III overexpression affects other synapses. While NRG1-III overexpression did not alter overall synapse density, we observed an increase in excitatory synapses. This finding aligns with previous reports of unbalanced excitatory-inhibitory neurotransmission in the cortex following increased NRG1-III expression [23] and is consistent with heightened motor activity in TG mice. NRG1-III overexpression in TG mice also occurs in dorsal root ganglion (DRG) neurons [21], which project glutamatergic afferents to MNs. Therefore, the altered number of glutamatergic synapses on MNs may result from elevated NRG1-III levels in sensory neurons, rather than from MN intrinsic changes in MNs. However, the mechanisms regulating the formation of excitatory inputs on MNs may be more complex, since increased VGlut-1-positive synapses occur in vitro in the absence of DRG neurons. Excitatory–inhibitory imbalance is a pathogenic mechanism underlying various neurodegenerative conditions. Overactivation of glutamate receptors on MNs in TG animals likely contributes to oxidative stress, disruption of Ca^2+^ homeostasis, and excitotoxicity, thereby adding another detrimental factor to the pathogenic framework operating in the NRG1-III overexpression model, as is also seen in MN diseases [56].

CGRP expression is upregulated during development in MN somata and NMJs, and increases in response to injury in adulthood [36,37,38]. Elevated CGRP levels observed in the MN somata of adult TG mice suggest a persistent state of neuromuscular system remodeling. However, the age-related increase in CGRP observed in WT mice [39] was not observed in TG mice, indicating impaired adaptation to age-dependent plastic changes. Additionally, the expression of MMP9, a marker of fast, vulnerability-prone MNs, also declines in aged TG mice. Since MMP9 is also upregulated during axonal regeneration [57], the dynamic changes in its expression can be interpreted in a similar way to those observed for CGRP. Furthermore, elevated BiP expression revealed increased ER stress in adult TG mice, supporting the notion that NRG1-III overexpression has detrimental effects on MNs by disrupting ER homeostasis, which undergoes significant structural rearrangements.

Our previous findings showed an increased density of microglial cells in the ventral horn of NRG1-III overexpressor animals [16]. This aligns with existing evidence of ErbB receptor expression in microglia and the ability of NRG1 to stimulate microglial activation [58]. It is conceivable that the basal overactivation of microglia observed in TG animals promotes a chronic pro-inflammatory state, potentially contributing to further neuronal damage. Peripheral nerve injury induces axonal repair processes in MNs [16,44,45]. C-boutons are disrupted in axotomized MNs, coinciding with selective microglial recruitment [16]. In the present study, increased NRG1-III expression further enhanced microglial recruitment to the soma of axotomized MNs. Moreover, the NRG1-labeled SSC, which was markedly enlarged in TG mice, became disrupted following axotomy. Notably, elevated microglial responses did not result in additional elimination of presynaptic components of C-boutons in TG animals. These findings suggest that the disruption of the expanded SSC may be part of the extensive ER reorganization associated with the chromatolytic response [59].

Several studies have established a link between the NRG/ErbB4 pathway and NMDAR signaling in cortical neurons [46,60,61,62]. For example, NRG2 activates ErbB4 and promotes its association with GluN2B-containing NMDARs which, in turn, leads to their downregulation [46]. In our study, MNs from TG animals were protected from NMDA-mediated excitotoxicity, possibly due to reduced expression of NMDARs as observed in cultures via immunocytochemistry. In line with this, recombinant human NRG1 enhances MN survival in spinal cord organotypic cultures subjected to chronic excitotoxicity [30].

Synaptic neurotransmission via C-boutons involves Ca^2+^ mobilization from the SSC. Additionally, Ca^2+^-dependent potassium channels at the postsynapse of C-boutons regulate MN excitability [8]. Given that NRG1-III overexpression promotes the formation of SSC-like ER–plasma membrane contacts, we sought to analyze its impact on Ca^2+^ dynamics in MNs. Our findings reveal that TG MNs exhibit elevated basal Ca^2+^ levels and spontaneous Ca^2+^ transients at a higher frequency. These results align with an increased number of excitatory synapses both in vitro and in vivo, as well as enhanced motor performance in TG animals. In addition, the role of metabotropic glutamate receptors (mGluR) in regulating the spontaneous Ca^2+^ oscillations observed in TG MNs should be further evaluated. In this regard, Ca^2+^ mobilization from the ER through mGluR1/5 activation has been shown to be involved in the generation of Ca^2+^ waves [63,64]. Elevated basal Ca^2+^ signals in TG MNs may render these cells more susceptible to various stressors, including aging-dependent axonal degeneration and SOD1-mediated MN disease [32,39].

NRG1-III plays a pivotal role in peripheral synapse formation and function. It mediates signaling at the NMJ through ErbB receptors which are expressed in pre- and postsynaptic terminals as well as in Schwann cells [65,66,67]. In null NRG1-III mutant mice, initial synapse formation during embryogenesis is followed by denervation and loss of tSC, ultimately leading to embryonic death at birth [68]. ErbB2-deficient mice exhibit pre- and postsynaptic defects at developing NMJs [69]. Moreover, previous studies have shown that NRG1-III overexpression accelerates the loss of motor inputs during developmental synapse elimination and enhances NMJ remodeling at developing and adult NMJs [22]. Our findings reveal that the NMJ phenotype in adult mice overexpressing NRG1-III mirrors the characteristics typically observed in aged NMJs, including increased fragmentation and partial denervation. Notably, the impact of NRG1-III overexpression appears to be potentiated with aging. Moreover, partial denervation was confirmed by electron microscopy, which revealed aged TG NMJs retaining postsynaptic folds in the absence of presynaptic terminals. Partial denervation has been previously described at NMJs from aged mice and in neuromuscular diseases such as ALS [39,70,71]. The increased number of degenerative NMJs aligns with our ultrastructural findings in aged TG mice, showing a prominent accumulation of autophagic structures in degenerating nerve terminals that were not reported in younger ages [22], suggests an intrinsic MN pathology during prolonged overexpression of NRG1-III. Autophagy is an increasingly recognized mechanism in the maintenance of normal synaptic function, also playing a relevant role in aging and neurodegenerative diseases [72]. At the NMJ, both metabolic and neuronal stimulation induce the formation of presynaptic autophagosomes [73]. The robust accumulation of autophagic vesicles observed in some degenerating motor nerve terminals of TG animals can be interpreted as a consequence of an imbalance in the activity-dependent upregulation of autophagy. An initial adaptive response may ultimately lead to the accumulation of autophagic vesicles due to impaired lysosomal clearance and reduced autophagic flux. Thus, a homeostatic response may become harmful over time. A similar autophagic vesicle overloading has been observed in degenerating presynaptic terminals in the brains of Alzheimer’s disease patients [74,75]. The metabolic pathway involving the TOR kinase, which has been implicated in aging, may also play a role in regulating autophagy at motor nerve terminals [76]. Moreover, although we confirmed an increase in tSCs at adult TG NMJs [22], we found that this proliferative effect was not maintained in aged mice.

The neuropeptide CGRP is transported along axons and released at NMJs [35,36]. CGRP levels at NMJs increase during development, but decline in adulthood, with a subsequent upregulation correlating with plastic changes during aging or following neuromuscular injury [37,38,39]. In adult NRG1-III TG mice, we observed elevated CGRP levels at NMJs, accompanied by structural changes indicative of synaptic plasticity. This finding agrees with previous reports of NMJ instability during adult stages, when NRG1-III is overexpressed in motor axons [22]. The hyperactivity of tSCs in response to NRG1-III overexpression may contribute to this alteration. GAP-43, a key protein involved in axonal growth, was also upregulated at TG NMJs in SOL, which additionally showed less degeneration compared with TA NMJs. However, degenerated NMJs did not exhibit increased CGRP levels in aged TG mice. This suggests that reparative plastic responses are insufficient to counteract the progressive degenerative processes characteristic of aging muscles. tSCs are essential players in NMJ plasticity, and a key molecular mediator of this process is NRG1 type III [22]. Our data showing tSC activation in adult but not in old NMJs are consistent with the CGRP expression dynamics observed.

## 4. Materials and Methods

### 4.1. Mice Colony and Animal Facilities

We used TG mice overexpressing NRG1-III: *C57Bl6*-Tg (Thy1-Nrg1*III) 1Kan^+/−^, which overexpress an N-terminally HA epitope-tagged NRG1-III isoform under the control of the neuronal Thy1.2 promoter [21]. Mice were housed in the Animal Facility of the Universitat de Lleida, maintained in a strictly controlled environment (12-h light/dark cycle and 20 ± 2 °C of room temperature [RT]) with ad libitum access to chow and water. Genomic DNA was extracted from tail biopsies at P0 using the Phire Kit (ThermoFisher Scientific, Waltham, MA, USA). The genotype of the offspring was determined by PCR with the following primers: 5′-GGCTTTCTCTGAGTGGCAAAGGACC-3′ (forward, HANI-Nrg1 transgene), and 5′-GTCCACAAATACCCACTTTAGGCCAGC-3′ (reverse, HANI-Nrg1 transgene).

All animal procedures were carried out in accordance with the European Committee Council Directive, Animal Care and Use and Biosecurity Committees of the Universitat de Lleida, and the norms established by the Generalitat de Catalunya (Diari Oficial de la Generalitat de Catalunya [DOGC] 2073, 1995).

Surgical manipulations were performed under anesthesia, using a combination of ketamine (100 mg/kg) and xylazine (10 mg/kg).

### 4.2. Motor Behavior Analysis

Behavioral assessments were conducted monthly from P330 until animal death (*n* = 8–24 mice in almost all time points). Lifespan data were also recorded. To evaluate motor function, the Open-field and the Catwalk tests were performed. Briefly, in the Open-field test, mice were placed on a platform, and their spontaneous motor activity was recorded for 5 min. The total distance covered and speed were analyzed using Smart software (v2.521, Panlab Harvard Apparatus, Spain). For the CatWalk XT test (Noldus, Wageningen, The Netherlands), mice were placed on an enclosed walkway across a glass plate and allowed to cross from one side to the other. Gait data were recorded and analyzed with CatWalk XT 10.0 software.

### 4.3. Nerve Transection Experiments

Adult mice (P60) were subjected to unilateral sciatic nerve transection. The animals were anaesthetized with a combination of ketamine (100 mg/kg) and xylazine (10 mg/kg). The sciatic nerve was exposed at the femoral level, transected, and a ligature was placed on the proximal segment to prevent spontaneous reinnervation. Postoperative analgesia was provided through two subcutaneous injections of buprenorphine (0.05 mg/kg), one immediately after surgery and the other 24 h later. Lumbar spinal cord samples were collected 7 days post-axotomy.

### 4.4. Spinal Cord Cultures

Primary cultures of dissociated spinal cord cells from WT and NRG1-III mice (embryonic day 13) were performed as previously described [77], with minor modifications. Briefly, lumbar spinal cords were dissected, and the meninges and ganglia were removed. After dissociation, cells were plated at a density of 300,000 cells per well on round coverslips coated with a poly-D-lysine plus Matrigel basement membrane matrix (Corning, Bedford, MA, USA). The cells were maintained in essential medium (Gibco, Waltham, MA, USA) enriched with 5 g/L glucose, 3% horse serum, 10 ng/mL nerve growth factor, and B27 medium (Gibco). On day 6, cultures were treated with 1.4 µg/mL cytosine-β-arabinoside (Sigma-Aldrich, Saint Louis, MO, USA) to inhibit non-neuronal cell growth. Cells were maintained for 14–23 DIV, then washed in phosphate-buffered saline (PBS), fixed in 4% paraformaldehyde (PFA) in 0.1 M phosphate buffer (PB) (pH 7.4) for 1 h, and processed for immunofluorescence. In some cultures, MN survival was assessed after treatment by applying to the cultures 100 µM NMDA (Sigma-Aldrich) for 30 min.

### 4.5. Tissue Preparation, Immunohistochemistry, and Image Analysis for Confocal Microscopy

Adult (P60–P90) and old mice (P530–P740), as well as axotomized mice (7 days post-surgery), were deeply anesthetized and transcardially perfused with physiological saline solution followed by 4% PFA in 0.1 M PB at pH 7.4. Lumbar spinal cords, SOL and TA muscles were dissected and post-fixed, either overnight (spinal cords) or for 2 h (muscles) at 4 °C, in the same fixative solution. Tissues were cryoprotected at 4 °C with 30% sucrose in 0.1 M PB containing 0.02% sodium azide. Transverse cryostat sections of spinal cords or longitudinal sections of muscles (16 µm thick) were collected on gelatine-coated glass slides.

Cryostat sections and coverslips with cultures were permeabilized with PBS containing 0.1% Triton X-100 for 30 min, blocked with either 10% normal goat serum or normal horse serum in PBS for 1 h at RT, and incubated overnight at 4 °C with a mixture of primary antibodies (Table 1). After primary antibody incubation, sections were washed and incubated for 1 h at RT with appropriate secondary fluorescent antibodies (1:500) labeled with one of the following fluorochromes: Alexa Fluor 488, DyLight 549 or DyLight 649 (Jackson Immuno Research Laboratories, West Grove, PA, USA). Spinal cord sections were then labeled with the blue fluorescent NeuroTrace Nissl staining (1:150, Molecular Probes). Muscle sections were incubated with Alexa Fluor 555 conjugated α-Bgtx (1:500, Molecular Probes) to label AChR. Some cultures and muscle samples were also stained with DAPI (50ng/mL, Molecular Probes) for DNA labeling. Samples were mounted using an anti-fading medium containing 0.1 M Tris-HCl buffer (pH 8.5), 20% glycerol, 10% Mowiol, and 0.1% 1,4-diazabicyclo[2,2,2]octane. Slides were examined using an Olympus BX51 epifluorescence microscope (Olympus, Hamburg, Germany) equipped with a DP30BW camera or, alternatively, with a laser-scanning confocal microscope (FluoView FV-1000 Olympus (Olympus, Hamburg, Germany) or a TCS SPE Leica [Leica Microsystems, Wetzlar, Germany]).

Spinal cord and muscle cryosections, as well as some culture samples, were imaged by obtaining optical sections (1 μm). Slides from different animals and experimental conditions were processed in parallel to immunocytochemistry and subsequent imaging. The same scanning parameters were used across different experimental groups to ensure consistency. Digital images were analyzed using FV10-ASW 3.1 Viewer (Olympus, Tokyo, Japan) and Fiji software https://imagej.net/imagej-wiki-static/Fiji (accessed on 1 June 2021) (US National Institutes of Health, Bethesda, MD, USA).

The area and perimeter of MN somata, as well as microglial profiles located close to MNs, were manually measured, respectively, on Nissl or Iba1-labeled images. Only MNs with a large nucleus and visible nucleolus were included in this analysis. Immunolabeled profiles of NRG1, VAChT, VGluT1, VGluT2, and VGAT, were manually counted or measured using the plot profile plugin in ImageJ (https://imagej.net). Immunoreactivity levels for CGRP and MMP9 in spinal cord sections, or NMDA receptor GluN2B and AMPA receptor GluA2 in cultured MNs, were quantified in digital images after background subtraction using ImageJ software. MN survival in some cultures was analyzed by manually counting ChAT-immunolabeled MNs, using ChAT positivity for their identification.

The cytoarchitecture of the NMJs was assessed in images projected from reconstructed Z-stack optical sections (1 μm). For immunofluorescence quantifications, the same number of optical sections was projected for all measurements. NMJ size was evaluated by determining the area and perimeter of manually outlined α-Bgtx-labeled whole postsynaptic sites. Additionally, the postsynaptic area enriched in clustered AChRs was measured. Extrasynaptic α-Bgtx-labeled spots were quantified in selected regions outside the NMJs. NMJs with a discontinuous appearance, characterized by five or more small islands of clustered AChR, were considered fragmented. The percentage of fragmented NMJs and the number of fragments per NMJ were calculated. Partial denervation was defined when the postsynaptic site was not fully aligned with the nerve terminal, which was immunostained for both NF68 and SV2 (as presynaptic markers). In muscles from old mice, NMJs with morphological signs of degeneration were counted. tSC in NMJs were quantified using S100 immunolabeling (a marker of tSC) combined with DAPI to identify tSC nuclei. Immunostaining for CGRP and GAP-43 was assessed based on pixel intensity after background subtraction in NMJs, delimited by α-Bgtx labeling.

The digital images were edited using FV10-ASW 3.1 Viewer (Olympus), ImageJ, and Adobe Photoshop CS4 (Adobe Systems Inc., San Jose, CA, USA).

### 4.6. Electron Microscopy and Ventral Nerve Root (VR) Analysis

Animals were perfused with 2% PFA and 2% glutaraldehyde in 0.1 M PB (pH 7.4). Lumbar spinal cords, L4 VR, and TA and SOL muscles were dissected and post-fixed for 24 h at 4 °C in the same fixative solution. Spinal cords were sectioned at 300 µm using a vibratome. The tissues were then post-fixed in 1% OsO4 for 2 h and processed for Embed 812 epoxy resin (Electron Microscopy Sciences, Hatfield, PA, USA), according to standard procedures. Ultrathin sections were counterstained with uranyl acetate and Reynold’s lead citrate. Observations were made using a JEOL JEM 1010 transmission electron microscope (Akishima, Tokyo, Japan).

Semithin sections (1 µm thick) of VR were stained with Richardson’s stain and imaged using an Olympus 60x/1.4NA PlanApo oil immersion objective (Olympus) and a Nikon DMX 1200 digital camera (Tokyo, Japan). The degree of myelination relative to axon cross-sectional size (G-ratio) was evaluated using the specific plugin in ImageJ, which computes the ratio between the inner axonal diameter and the total diameter (axon plus the surrounding myelin sheath).

### 4.7. Calcium Imaging

For Ca^2+^ imaging experiments, primary spinal cord cultures were maintained for 14–23 DIV. The cultures were loaded for 3 min at 37 °C with the Ca^2+^ indicator Fura-2 acetoxymethyl ester (Fura-2 AM, 5 µM, Molecular Probes), dissolved in a perfusion solution containing (in mM): 150 NaCl, 5 KCl, 1 CaCl_2_, 1 MgCl_2_, 20 Hepes buffer (pH 7.4), 10 glucose, and 1% bovine serum albumin. After washing in perfusion buffer, cells were incubated in the buffer for 15 min at RT for Fura-2-AM de-esterification. The slices were then transferred to a perfusion chamber PH1 (Warner Instrument, Handem, CT, USA) and placed in a Nikon Eclipse TE200 inverted fluorescence microscope equipped with a Spectramaster monochromator and an Orca ER camera (Hamamatsu Photonics, Hamamatsu City, Japan). The system was controlled by the Aquacosmos 2.6 software (Hamamatsu Photonics) for Ca^2+^ imaging. The following agents were administered directly into the recording chamber or delivered through perfusion with the previously mentioned solution: 100 µM NMDA, 100 µM KA, (Sigma-Aldrich), 50 mM KCl, 20 mM caffeine (Sigma-Aldrich), and 1 µM thapsigargine (Sigma-Aldrich). MNs in the cultures were identified based on their large size and shape. Dual excitation wavelengths (340 and 380 nm) were used, and ratio images were captured using a 40X 1.3 NA Nikon S Fluor oil-immersion objective and subsequently analyzed (Tokyo, Japan).

### 4.8. Western Blotting

Fresh frozen cells from the primary spinal cord cultures were processed for protein extraction using RIPA lysis buffer (50 mM Tris–HCl [pH 7.4], 150 mM NaCl, 1 mM EDTA, 1% NP-40, 1% Na-deoxycholate, 0.1% SDS) supplemented with protease inhibitors (Sigma-Aldrich, cat. # P8340) and phosphatase inhibitors (PhosSTOP, Roche, Laval, Canada cat. # 04906837001). Protein concentrations of the supernatants were determined by BIO-RAD Micro DC protein assay (BIO-RAD, Laboratories, Inc., Hercules, CA, USA). Forty µg of protein were loaded onto SDS-polyacrylamide gels (7.5–10%) and transferred to polyvinyldifluoride membranes (ImmobilonTM-P, Millipore cat. # IPVH00010). Membranes were blocked with 5% dried skim milk in 0.1% Tween and Tris-buffered saline (TBST, pH 8) for 1 h at RT, followed by washing in TBST. The membranes were then incubated overnight at 4 °C with primary antibodies (Table 1). After washing in TBST, membranes were incubated with the secondary antibodies: anti-rabbit IgG HRP-linked (1:20,000, Cell Signalling, cat. # Ref: 7074), anti-mouse IgG HRP-linked (1:20,000, Cell Signalling, cat. # Ref: 7076) for 1 h at RT. After further washing in TBST, protein bands were visualized using the ECL Prime Western blotting Detection Reagent detection kit (Amersham, UK, cat. # RPN2236). Imaging was performed using the Chemi-Doc MP Imaging System (BIO-RAD Laboratories Inc., Hercules, CA, USA), and densitometric analysis was carried out using Image Lab 4.0 software.

### 4.9. Statistical Analysis

Data are presented as means ± SEM. Statistical comparisons were made using either one- or two-way analysis of variance (ANOVA) followed by post hoc Bonferroni’s test, or Student’s *t*-test, as appropriate. Differences were considered statistically significant when *p* value was ≤0.05. GraphPad Prism 6 software was used for statistical analysis and graph representation of the data.

## 5. Conclusions

Our findings reveal that NRG1 overexpression triggers several chronic plastic changes in the neuromuscular system. These include altered organization of C-boutons and other MN afferents, changes in MN and glial responses to injury, and autophagic disruption of nerve terminals at NMJs. These detrimental outcomes should be carefully considered when developing therapeutic strategies involving NRG1 for the treatment of neuromuscular diseases.

## Figures and Tables

**Figure 1 ijms-26-11421-f001:**
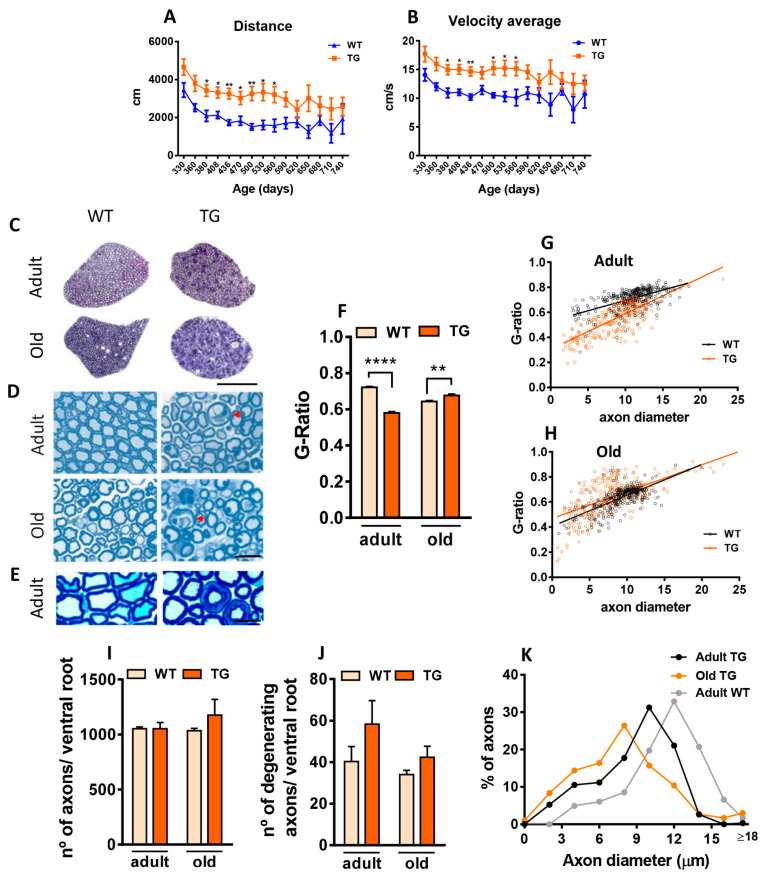
NRG type III (NRG1-III) overexpression increases motor activity and changes motor axon myelination. (**A**,**B**) Open-field analysis of motor performance in wild-type (WT) and transgenic (TG) mice. Data show elevations in the distance covered (**A**) and the average velocity (**B**) in TG mice. (**C**–**I**) Examination of L4 ventral nerve root (VR) axons from adult and aged mice using transverse semithin plastic sections. (**C**–**E**) Representative micrographs of semithin sections from L4 VRs illustrate increased myelin thickness in adult TG axons and the presence of degenerating axons (arrows) in both adult and old TG mice. (**F**–**H**) G-ratio measurements indicate increased myelin thickness in adult TG mice and decreased thickness in aged TG mice. (**I**) Quantification of axon numbers in VRs reveals no differences between WT and TG mice in both adult and aged groups. (**J**) A trend toward a higher number of degenerating axons was detected in TG mice in both age groups (adult: *p* = 0.25; old: *p* = 0.22). (**K**) Relative frequency histogram of axon diameters for adult and old TG, and adult WT L4 VR axons; note the increased proportion of small-caliber axons in old TG, as well as the appearance of a small population of very large axons, probably representing those with age-related swelling and degenerative changes. For motor behavior analyses, sample sizes ranged from *n* = 8–24 per time point (one WT point at *n* = 2 and occasionally WT points at *n* = 4). * *p* < 0.05, ** *p* < 0.01 (Student’s *t*-test for genotype comparisons at each time point). For ventral root axons measurements, sample sizes were as follows: g-ratio *n* = 204–275 axons from 3 animals; axon number and degenerating axon counts, *n* = 3 animals per condition (Student’s *t*-test [for genotype comparisons] and two-way ANOVA, Bonferroni‘s post hoc test). ** *p* < 0.01 and **** *p* < 0.0001. Scale bars: 200 µm in (**C**), 50 µm in (**D**), 15 µm in (**E**).

**Figure 2 ijms-26-11421-f002:**
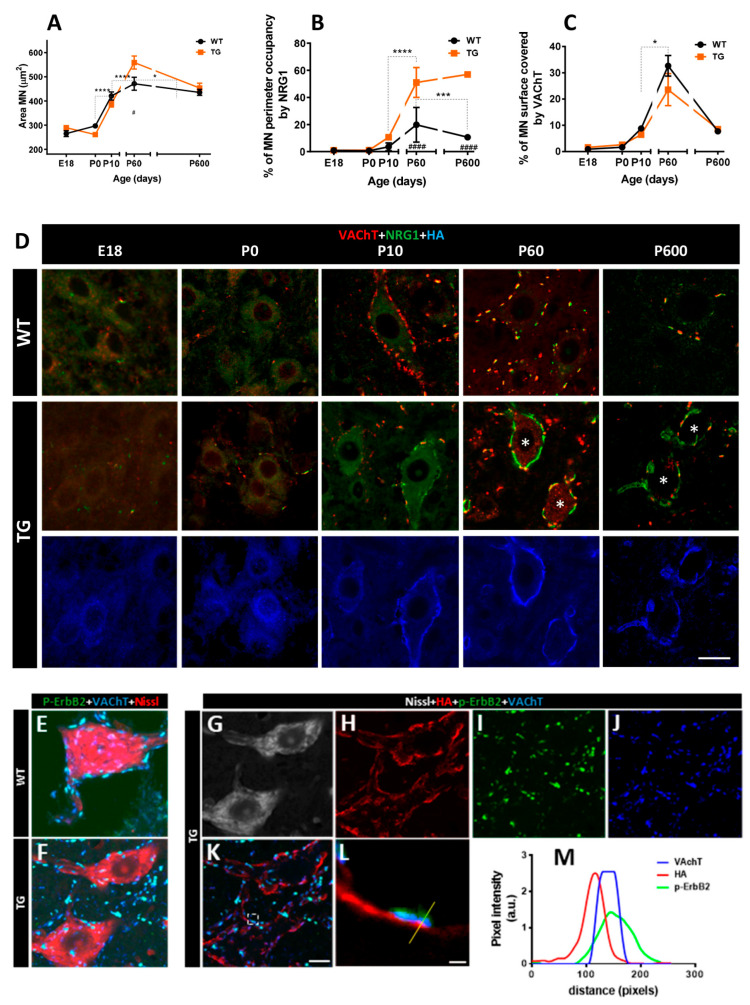
Impact of NRG1-III overexpression on C-bouton organization. Graphs depict the time course of the average motor neuron (MN) area (**A**), and the MN perimeter occupied by NRG1 (**B**) and vesicular acetylcholine transporter (VAChT) (**C**). (**D**) Representative confocal micrographs from each time point show MNs immunolabeled for VAChT (red), NRG1 (green), and hemagglutinin (HA) (blue), illustrating an expansion of NRG1 near the surface of TG MNs at P60 and P600 (*), while VAChT remains unchanged. (**E**–**L**) MNs labeled for HA (red), p-ErbB2 (green), VAChT (blue) and counterstained with fluorescence Nissl staining (grey), showing the presynaptic association of ErbB2 with VAChT facing HA postsynaptic labeling in both WT and TG MNs. An enlarged detail of the region outlined in panel (**K**) is provided in panel (**L**). (**M**) Pixel profile analysis across the line indicated in panel (**L**) confirms an overlap between VAChT and p-ErbB2 signals but reveals a shift in the peak intensities of VAChT and p-ErbB2 relative to HA. Simple sizes in (**A**) ranged from *n* = 20–104 MNs from 3 mice; in (**B**,**C**) *n* = 21–41 MNs from 3 mice. * *p* < 0.05, *** *p* < 0.001, **** *p* < 0.0001, # *p* < 0.05 and #### *p* < 0.0001. Scale bars: 20 µm in (**D**), 10 µm in (**E**–**K**), and 1 µm in (**L**).

**Figure 3 ijms-26-11421-f003:**
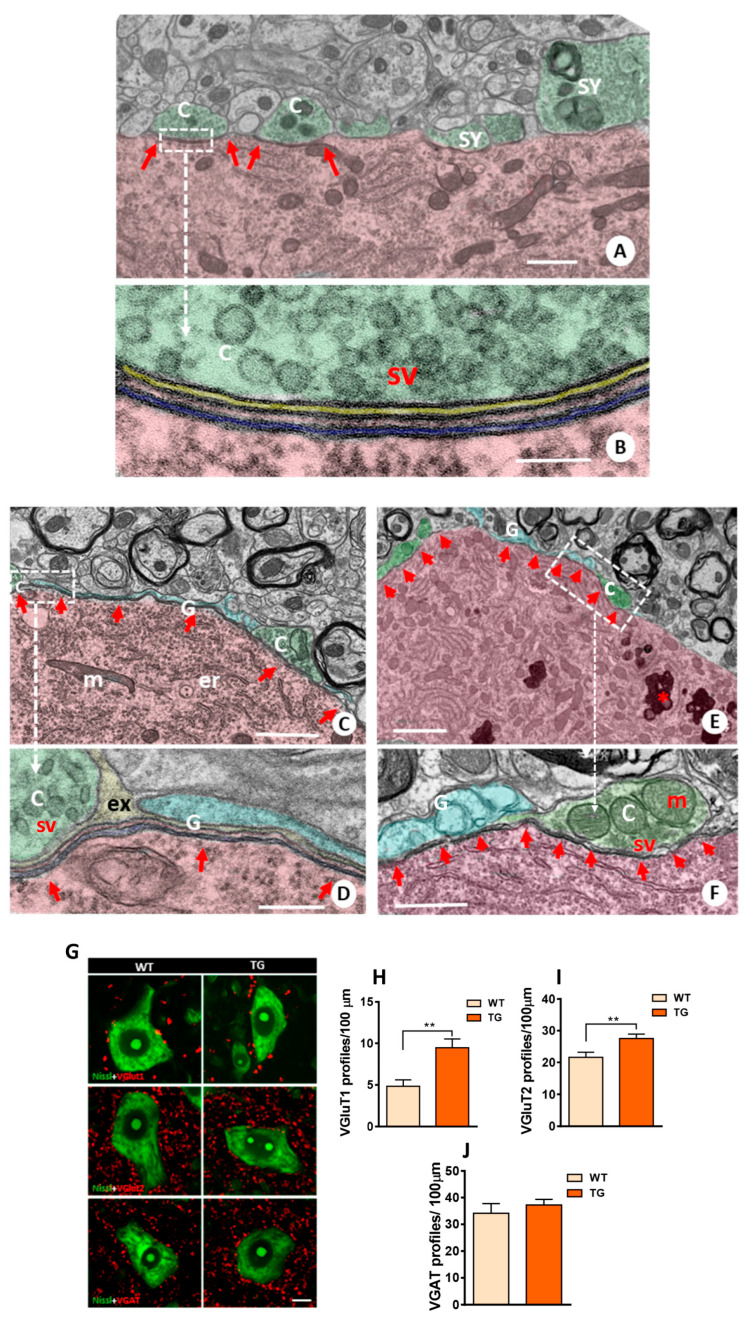
Structural changes in MN afferent synapse organization induced by NRG1-III overexpression. (**A**) WT MN in adult mice showing C-boutons (**C**) and additional synapses (SY) contacting MN surface in adult mice (P100); note that the subsynaptic cistern (SSC) is restricted to C-bouton postsynaptic region (arrows); presynaptic terminals are shaded in green and MN soma is colored in red. (**B**) An enlarged view of the area outlined in (**A**) detailing the particular organization of C-bouton membrane compartmentation in WT mice in which extracellular space (yellow) and postsynaptic SSC (blue) are delimitated; presynaptic C-bouton terminal with clear spherical vesicles and postsynaptic neuron are shaded in green and red, respectively. (**C**) MN from TG mice (P41) show a prominent enlargement of the SSC adjacent to the plasma membrane (arrows) outpacing the limits of presynaptic C-bouton terminals in which SSC remains circumscribed in WT; additionally, glial processes (**G**) are observed covering part of MN surface adjacent to SSC but devoid of synapses. (**D**) A further enlargement of the region marked in (**C**), highlighting the SSC growth induced by NRG1-III overexpression; the color code is the same as A and B with additional shading of glial processes in clear blue. (**E**,**F**) MN from old TG mice (P725) showing the similar organization of C-boutons and expanded SSC (arrows) as observed in adults. The color code is the same as (**A**–**D**). Note the accumulation of lipofuscin granules seen in old MNs (*). sv = synaptic vesicles; m = mitochondria; er = endoplasmic reticulum; ex = extracellular space. (**G**–**J**) Changes in afferent synaptic boutons other than C-boutons on MNs: MN somata labeled for vesicular glutamate transporter 1 (VGluT1), vesicular glutamate transporter 2 (VGluT2), and vesicular GABA transporter (VGAT) (red) and Nissl (green); the distribution of each synaptic type on MN surface is shown (**G**) and quantified in (**H**–**J**). Sample sizes in VGluT1 and VGluT2 graphs ranged from *n* = 15–17 MNs, and VGAT *n* = 18 MNs per condition from 2 animals. ** *p* < 0.01 (Student’s *t*-test for genotype comparisons). Scale bars: 1 µm in (**A**), (**C**) and (**F**), 200 nm in (**B**), 250 nm in (**D**), 2.5 µm (**E**) and 10 µm in (**G**).

**Figure 4 ijms-26-11421-f004:**
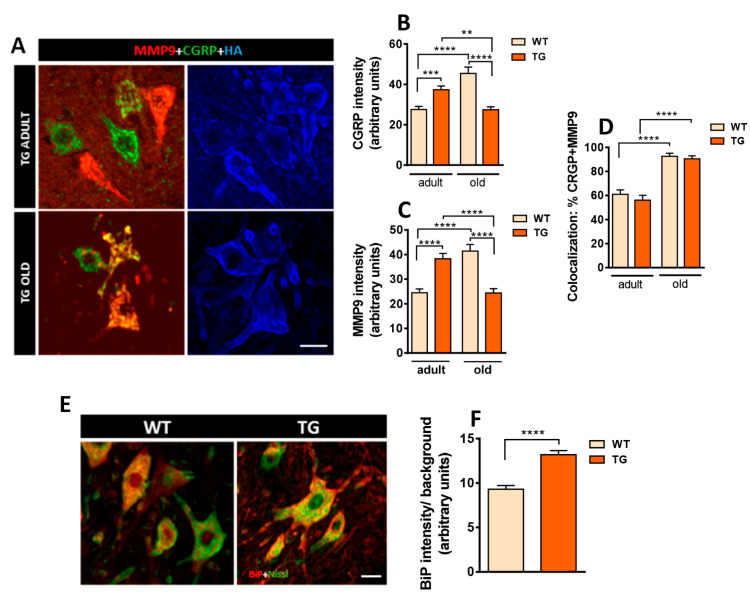
Calcitonin gene-related peptide (CGRP), matrix metalloproteinase-9 (MMP9), and immunoglobulin heavy chain binding protein (BiP) immunoreactivity in MNs of WT and TG mice. (**A**) Immunostaining for MMP9 (red), CGRP (green), and HA (blue) in spinal cord ventral horn of adult and aged TG mice; note the colocalization of MMP9 and CGRP in some MNs. (**B**,**C**) Age-related changes in the intensity of MMP9 and CGRP immunoreaction in WT and TG MNs. (**D**) Quantification of the proportion of MNs double-labeled for MMP9 and CGRP. (**E**,**F**) Immunolabeling and quantification of BiP (red) in MNs delimited by fluorescent Nissl staining (green). For CRGP intensity quantification, sample sizes ranged from *n* = 96–165 MNs from 3 animals; for MMP9 *n* = 165–242 MNs from 3 animals; for the percentage of CGRP-MMP9 colocalization, *n* = 18–31 images; for BiP *n* = 21–29 MNs from 2 animals (one-way ANOVA and Bonferroni‘s post hoc test, and Student’s *t*-test). ** *p* < 0.01, *** *p* < 0.001 and **** *p* < 0.0001. Scale bars: 25 µm in (**A**) and 20 µm in (**E**).

**Figure 5 ijms-26-11421-f005:**
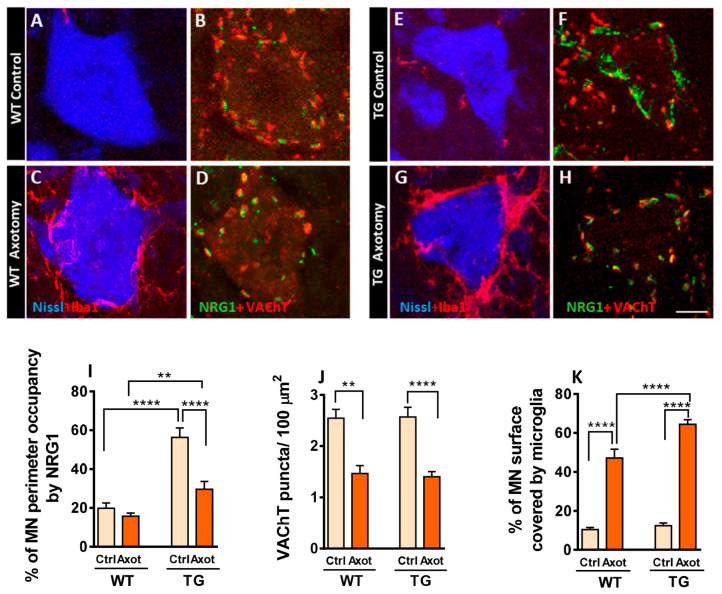
Impact of NRG1-III overproduction on the MN cell body response and central glial reaction to peripheral nerve injury. In (**A**,**C**,**E**,**G**) WT and TG MNs stained with Nissl (blue) show microglia (red) contacting MN surface, highlighting an exacerbated microglial response in TG MNs at P60, 7 days post-axotomy; in (**B**,**D**,**F**,**H**) double labeling for NRG1 (green) and VAChT (red), showing the reduction in C-bouton synapses after axotomy in both WT and TG MNs and the notable decrease in peripherally associated NRG1 in the soma of axotomized TG MNs. Graphs depict measurements of the perisomatic–microglial covering of MN (**I**), the density of C-boutons revealing that the reduction following axotomy is not altered in TG MNs (**J**), and the proportion of the MN periphery exhibiting NRG1 labeling (**K**). Ctrl (control), Axot (axotomy). Sample sizes in (**I**) ranged from *n* = 13–48 MNs, in (**J**) *n* = 10–39 MNs and (**K**) *n* = 10–31 MNs from 3 animals per condition. One-way and two-way ANOVA, Bonferroni‘s post hoc test). ** *p* < 0.01 and **** *p* < 0.0001. Scale bar: 10 µm (**A**–**H**).

**Figure 6 ijms-26-11421-f006:**
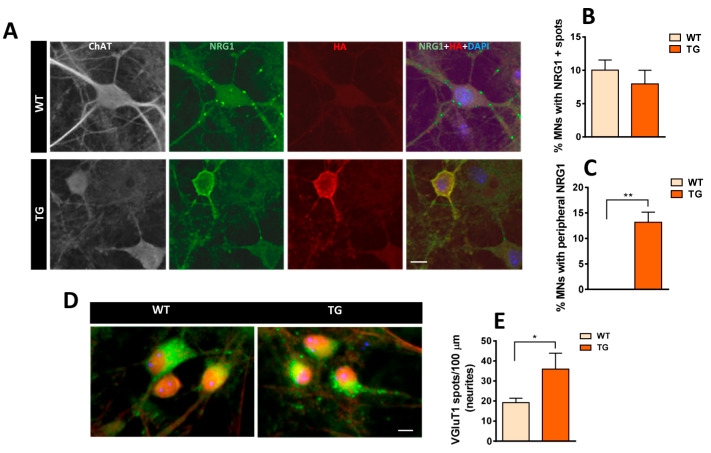
Immunolocalization of NRG1 expression along with VGluT1 to mark excitatory afferent synapses developed in cultured spinal cord MNs from WT and TG mice (19 days in vitro [DIV]). (**A**) MNs showing NRG1 (green), HA (red), choline acetyltransferase (ChAT) (grey), and 4′,6-diamidino-2-phenylindole dihydrochloride (DAPI) (blue). In WT MNs, NRG1 appears as scattered spots near the cell surface, whereas in TG MNs expressing HA, a pronounced peripheral accumulation is observed. Graphs illustrate the percentage of MNs exhibiting either a spotty pattern (**B**) or peripheral accumulation (**C**) of NRG1 distribution. (**D**) Immunostaining for VGluT1 in MNs from WT and TG cultures, synaptic spots on neurites are quantified in (**E**). For the percentage of MNs with NRG1 positive spots and peripheral accumulation of NRG1 *n* = 3 cultures; for VGluT1 density *n* = 23–29 neurites from 3 cultures. * *p* < 0.05, ** *p* < 0.01 (Student’s *t*-test for genotype comparisons). Scale bars: 10 µm.

**Figure 7 ijms-26-11421-f007:**
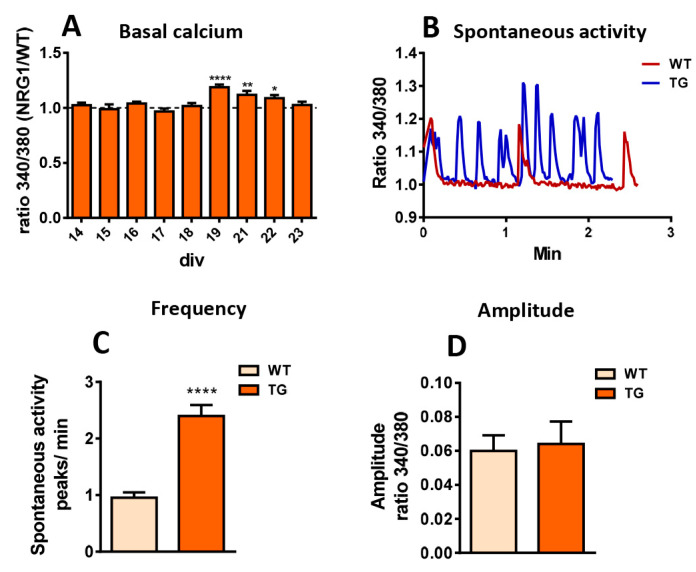
Calcium homeostasis in MNs from mice overexpressing NRG1-III. (**A**) Ca^2+^ recordings from cultured MNs (14–23 DIV) reveal increased basal Ca^2+^ levels in TG compared to WT. (**B**) Representative profile of intracellular Ca^2+^ dynamics in basal conditions of WT and TG cultures illustrating spontaneous activity (18 DIV). (**C**,**D**) Quantitative analysis of Ca^2+^ transient frequency (**C**) and amplitude (**D**) in cultures (14, 15, 16, 18 and 19 DIV). Sample sizes for basal Ca^2+^ measurements ranged from *n* = 21–230 MNs and for spontaneous activity frequency and amplitude from *n* = 41–49 cultures, 176–235 MNs. * *p* < 0.05, ** *p* < 0.01, **** *p* < 0.0001 (Student’s *t*-test for genotype comparisons).

**Figure 8 ijms-26-11421-f008:**
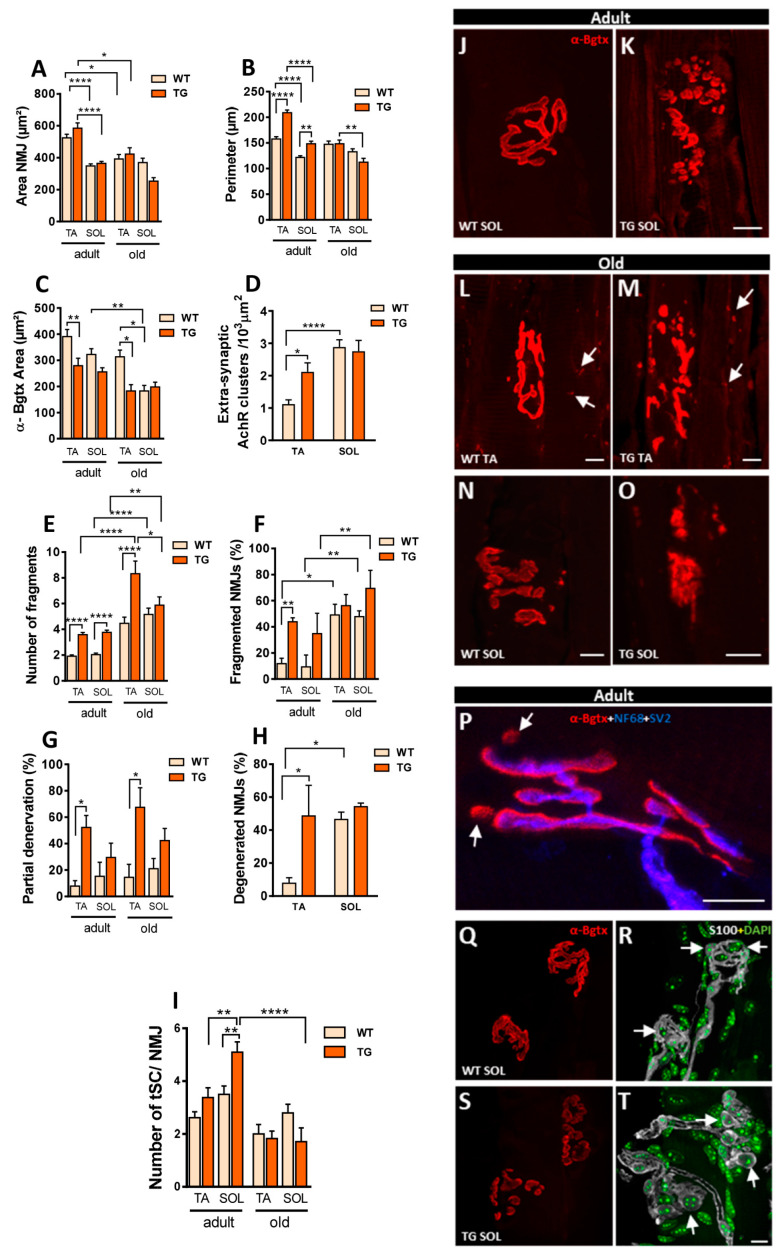
Neuromuscular junctions (NMJs) are altered in mice overexpressing NRG1-III. Changes in NMJ whole size (**A**), perimeter (**B**), and postsynaptic area delimited by α-Bungarotoxin (α-Bgtx) (**C**) in muscles from adult and old WT and TG mice. (**D**) Density of extrasynaptic α-Bgtx spots in soleus (SOL) and tibialis anterior (TA) muscles from aged mice. The number of NMJ fragments is shown in (**E**), and the percentage of fragmented NMJs in (**F**). The amount of partial denervated NMJs in adult and old muscles is displayed in (**G**) and the incidence of NMJs with degenerative morphology in WT and aged TG muscles in (**H**,**I**) Changes in terminal Schwann cell (tSC) numbers at NMJs in adult and old mice (WT and TG). (**J**–**O**) Representative images of adult and aged NMJs from WT and TG mice stained for α-Bgtx (red) illustrating various structural changes such as fragmentation (**K**,**M**,**O**); extrasynaptic acetylcholine receptors (AChR) spots indicated by arrows (**L**,**M**) and degeneration in TG (**O**) evidenced by a disrupted morphology when compared with WT (**N**,**P**) An NMJ immunostained for α-Bgtx (red) and 68 kDa neurofilament-L (NF68) plus synaptic vesicle glycoprotein 2A (SV2) (blue), showing postsynaptic domains lacking presynaptic labeling (arrows), indicative of partial denervation in the TG. (**Q–T**) NMJs labeled for S100 (gray) as a t-SC marker in conjunction with α-Bgtx (red) and DAPI (blue); arrows point to S100-positive cells at the NMJ. For quantification, simple sizes are: NMJ area and perimeter, *n* = 24–137 NMJs from 3–7 mice; α-Bgtx area *n* = 21–33 NMJs from 3 mice; extrasynaptic α-Bgtx spots, *n* = 24–43 NMJs from 3–4 mice; number of NMJ fragments, *n* = 24–45 NMJs from 3–4 mice; percentage of fragmented NMJs, *n* = 3–4 mice (24–45 NMJs per muscle); percentage of partial denervation, *n* = 3–4 mice (NMJs = 33–44 per muscle); number of tSC per NMJ, *n* = 14–23 NMJs from 2–3 mice (one-way or two-way ANOVA and Bonferroni’s post hoc test, and Student’s *t*-test). * *p* < 0.05, ** *p* < 0.01, and **** *p* < 0.0001. Scale bars: 10 µm.

**Figure 9 ijms-26-11421-f009:**
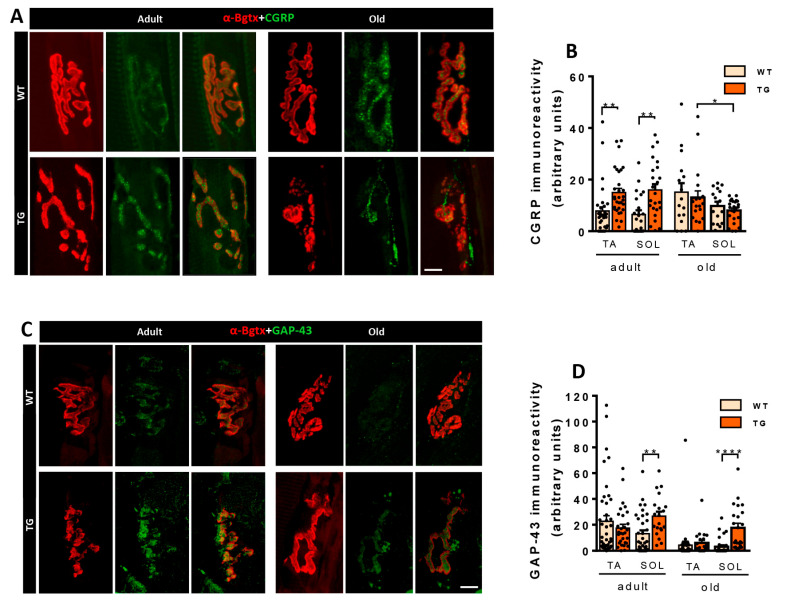
Proteins involved in the plasticity of NMJs, such as CGRP and growth-associated protein 43 (GAP-43), are upregulated in mice overexpressing NRG1-III. (**A**) Representative images showing CGRP immunoreactivity (green) at NMJs labeled with α-Bgtx (AChR, red) in (TA) muscles of adult and aged mice (WT and TG). Note that CGRP immunoreactivity is higher in NMJs of adult TG mice compared to WT, while no difference is observed in old mice. (**B**) Quantification of CGRP immunoreactivity at NMJs in TA and SOL muscles of adult and old mice (WT and TG). (**C**) Representative images of GAP-43 immunoreactivity (green) at NMJs labeled with α-Bgtx (AChR) (red) in SOL muscles of adult and aged mice (WT and TG) showing increased GAP-43 positivity in both adult and aged SOL muscles of TG mice. (**D**) The graph depicts measurements of GAP-43 immunolabeling at NMJs in TA and SOL muscles of adult and old mice (WT and TG). Simple size for CGRP intensity analysis ranges from *n* = 16–33 NMJs (3–4 animals), and for GAP-43 from *n* = 20–42 NMJs (3–4 animals). * *p* < 0.05, ** *p* < 0.01, **** *p* < 0.0001 (Student’s *t*-test for genotype comparisons). Scale bars: 10 µm.

**Figure 10 ijms-26-11421-f010:**
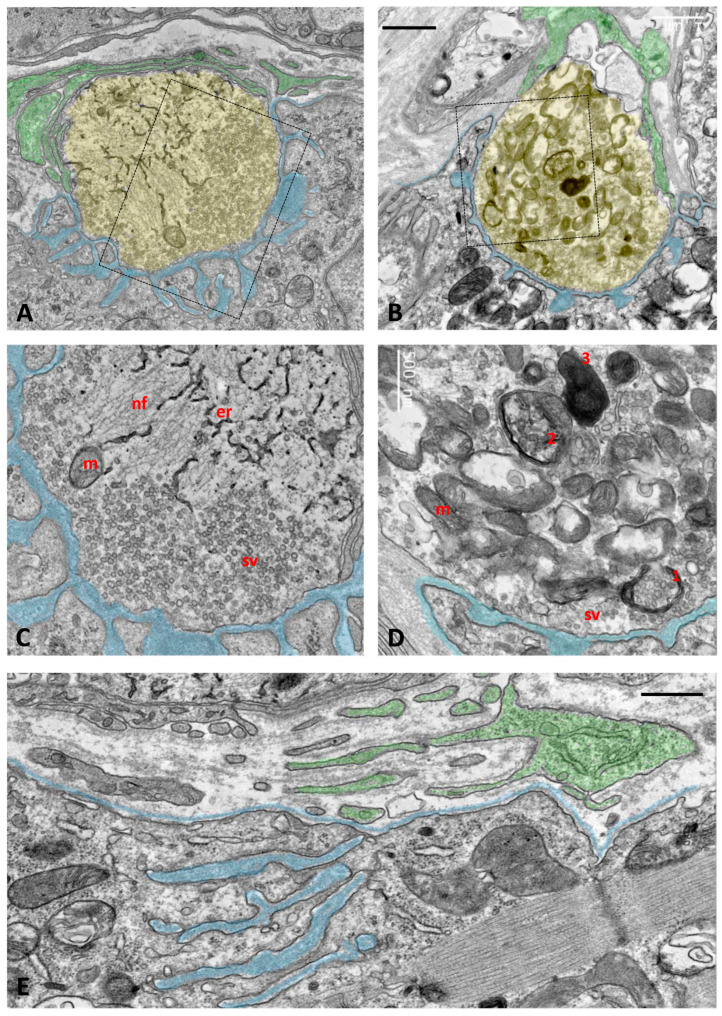
Ultrastructural analysis of NMJs of aged TG mice. Some nerve terminals appear morphologically normal, while others exhibit various degrees of degeneration. (**A**–**D**) Representative images of a normal and a degenerating nerve terminal, taken from the SOL and TA muscles of aged mice (P533), are shown in panels (**A**) and (**B**), respectively. Presynaptic terminals are shaded in yellow, postsynaptic membrane foldings are indicated in blue, and tSC profiles capping the nerve terminals are shaded in green. Enlarged views of the indicated regions are presented in panels (**C**) and (**D**); note the clustered synaptic vesicles (sv) near the presynaptic membrane, mitochondria (m), endoplasmic reticulum (er), and neurofilament bundles (nf). In panel (**D**), the degenerating nerve terminal displays an accumulation of autophagosomal–lysosomal-like structures at presumptive different stages of formation: (1) phagophore-like membranes engulfing clusters of synaptic vesicles, (2) autophagosomes containing synaptic vesicle remnants, and (3) electron-dense lysosomes. Several vacuolated mitochondria (vm) are observed adjacent to morphologically normal mitochondria (m). (**E**) A fully denervated NMJ is also shown: note the muscle fiber basal lamina and subneural folds (blue) lacking a facing presynaptic terminal; the whole structure is partially covered by tSC profiles (green). Scale bars: 1 µm in (**A**,**B**); 500 nm in (**C**–**E**).

**Table 1 ijms-26-11421-t001:** Primary antibodies used for immunocytochemistry and Western blotting.

Target	Source	Host Species	Used Concentration
CGRP	Sigma-Aldrich (St. Louis, MO, USA) (C-81989)	Rabbit polyclonal	1:1000
ChAT	MilliporeSigma (Temecula, CA, USA) (AB144)	Goat polyclonal	1:250
Glucose-regulated protein78 (Grp78/BiP)	Enzo Life Sciences (Farmingdale, NY, USA) (SPA-826)	Rabbit polyclonal	1:1000
GluR2 (GluA2)	Invitrogen (Waltham, MA, USA) (71-8600)	Mouse monoclonal	1:125
GAP-43	Novusbio (Centennial, CO, USA) (NB300-143)	Rabbit polyclonal	1:500
HA	Roche (Basel, Switzerland) (11 867 423 001)	Rat monoclonal	1:500
Ionised calcium-binding adaptor molecule 1 (Iba1)	Abcam (Cambridge, UK) (ab5076)	Goat polyclonal	1:500
Voltage gated potassium channel 2.1 (Kv2.1)	NeuroMAB (Davis, CA, USA) (73-014)	Mouse monoclonal	1:100
M2 muscarinic receptor (M2)	Alomone Labs (Jerusalem, Israel) (AMR-002)	Rabbit polyclonal	1:100
MMP9	Sigma-Aldrich (M9570)	Goat polyclonal	1:10
1α/β1/2 NRG1	Santa Cruz Biotechnology (Dallas, TX, USA) (sc-348)	Rabbit polyclonal	1:300
NF68	Abcam (ab72997)	Chicken polyclonal	1:1000
NMDAR1 (GluN1)	Invitrogen (PA5-34599)	Rabbit polyclonal	1:1000 (W)
NMDAR2A (GluN2A)	Invitrogen (480031)	Rabbit polyclonal	1:1000 (W)
NMDAR2B (GluN2B)	Invitrogen (71-8600)	Rabbit polyclonal	1:125 1:1000 (W)
phospho-Neu try1248 (p-ErbB2)	Santa Cruz Biotechnology (sc-293110)	Rabbit polyclonal	1:100
S100	Dako Agilent (Santa Clara, CA, USA) (Z311)	Rabbit polyclonal	1:500
SV2	Developmental Studies Hybridoma Bank (Iowa City, IA, USA) (SV2)	Mouse monoclonal	1:1000
Synaptophysin (SYN)	Synaptic Systems (Goettingen, Germany) (101 004)	Guinea pig polyclonal	1/500
Tubulin	Sigma-Aldrich (T5168)	Mouse monoclonal	1:20,000 (W)
VAChT	Synaptic Systems (139 105)	Guinea pig polyclonal	1:500
VGAT	Synaptic Systems (131 004)	Guinea pig polyclonal	1:200
VGluT1	Synaptic Systems (135 304)	Guinea pig polyclonal	1:500
VGluT2	Synaptic Systems (135 404)	Guinea pig polyclonal	1:500

## Data Availability

The datasets used and/or analysed are available from the corresponding author upon reasonable request.

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
