# Peer review of "Chronic Overexpression of Neuronal NRG1-III in Mice Causes Long-Term Detrimental Changes in Lower Motor Neurons, Neuromuscular Synapses and Motor Behaviour"

_ijms, 2025, doi:10.3390/ijms262311421_

Round 1
Reviewer 1 Report
Comments and Suggestions for Authors
- There is no definition of the “G-coefficient” in the caption to Fig. 1 and in the text.
- Panel M in Fig.2 show overlap of all 3 curves although authors claim “no overlap observed for HA“ . Also I failed to find definition of the “HA” in the legend to Fig.2, in the legends to subsequent figures, in the text and in the list of Abbreviations.
- The presence of spontaneous calcium spikes may indicate activation of Type I metabotropic glutamate receptors (probably mGluR5). The authors should test this, for example, using selective mGluRs agonists or by adding glutamate in the presence of NMDA and AMPA receptor inhibitors. At the very least, possible differences in the activation of the signaling pathway triggered by mGluRs in neurons of transgenic animals compared to controls should be considered (perhaps in the Discussion section).
Reviewer 2 Report
Comments and Suggestions for Authors
This work is characterized by an ingenious design, a huge volume of carefully and professionally executed methods, interesting results that are extremely important for understanding the mechanisms of action of neuregulins in nervous tissue.
The small number of comments listed below are not fundamental and do not detract from the high level of work.
- Abstract: It is necessary to reflect the methods and make the logic of the presentation clearer so that it is understandable not only to narrow specialists
- Considering the large number of abbreviations, it would be worth adding a separate list of them
- Lines 92-93: presumably part of the phrase was dropped out
- Figure 1 C: the image is too small; it might be worth adding it to the group of original images (if they will be available when published)
